# The impact of Neglected Tropical Diseases (NTDs) on health and wellbeing in sub-Saharan Africa (SSA): A case study of Kenya

**Elizabeth A. Ochola** [1]*, **Diana M. S. Karanja** [2], **Susan J. Elliott** [1]

**1** Department of Geography and Environmental Management, University of Waterloo Waterloo, Ontario, Canada, **2** Centre for Global Health Research, Kenya Medical Research Institute Kisumu, Kenya

☯ These authors contributed equally to this work.
* susan.elliott@uwaterloo.ca

**Data Availability Statement:** All relevant data are within the manuscript and its Supporting Information files.

## Abstract

Neglected Tropical Diseases (NTDs) remain endemic to many regions of sub-Saharan Africa (SSA) left behind by socioeconomic progress. As such, these diseases are markers of extreme poverty and inequity that are propagated by the political, economic, social, and cultural systems that affect health and wellbeing. As countries embrace and work towards achieving the Sustainable Development Goals (SDGs), the needs of such vulnerable populations need to be addressed in local and global arenas. The research uses primary qualitative data collected from five NTD endemic counties of Kenya: interviews key informants (n = 21) involved in NTD implementation programs and focus groups (n = 5) of affected individuals. Informed by theories of political ecology of health, the research focuses on post-devolution Kenya and identifies the political, economic, social, and cultural factors that propagate NTDs and their effects on health and wellbeing. Our findings indicate that structural factors such as competing political interests, health worker strikes, inadequate budgetary allocations, economic opportunity, marginalization, illiteracy, entrenched cultural norms and practices, poor access to water, sanitation and housing, all serve to propagate NTD transmission and subsequently affect the health and wellbeing of populations. As such, we recommend that post-devolution Kenya ensures local political, economic and socio-cultural structures are equitable, sensitive and responsive to the needs of all people. We also propose poverty alleviation through capacity building and empowerment as a means of tackling NTDs for sustained economic opportunity and productivity at the local and national level.

## Author summary

Wellbeing is currently seen as an avenue that shapes, happiness, productivity, environmental awareness, social inclusion, and justice; however, most countries presently adopting wellbeing measures are in the global north. Neglected Tropical Diseases (NTDs) significantly compromise populations' health and well-being in the global south, causing undesirable effects on the personal, social, and economic capabilities of communities living in endemic regions. As countries work towards achieving the Sustainable

**Funding:** We would like to thank the Queen Elizabeth scholarship Program and the David Johnston International Experience Award for the funding support. The funders had no role in study design, data collection and analysis, decision to publish, or preparation of the manuscript.

**Competing interests:** The authors have declared that no competing interests.

Development Goals (SDGs), it is paramount that countries in the global south adopt well-being measures and, in doing so, capture the lived experiences of individuals experiencing inequities, particularly as these are shaped by NTDs. In sub-Saharan Africa (SSA), political and economic power manifests across different scales, determining human-environmental interaction, distribution of resources, and the transmission of infectious agents. This paper uses a political ecology of health approach to identify the political, economic, social and cultural factors that contribute to NTDs, which are among the world's greatest global health problems. The vast majority of people bearing the burden of NTDs reside in low-income countries. Surprisingly, as the economies of the low-income countries improve to middle-income status, NTDs continue to thrive among sub-populations of low socioeconomic status because of the unequal distribution of economic gains. As a result, NTDs are found in environments characterized by poverty and income inequality, and other inequalities in access to health services, housing, safe water, and sanitation. This paper uses political ecology of health in the primarily biomedical field of NTDs to demonstrate that inequities are embedded within the broad political, socio-cultural, and economic systems that exacerbate NTD infection. The study recommends that NTD endemic countries in SSA formulate policies that enhance equity through capacity building and empowerment enhance population wellbeing.

## Introduction

In the last decade, sub-Saharan Africa (SSA) has experienced tremendous economic growth; for example, Foreign Direct Investments (FDI) in the region has tripled, making it the fastest-growing economy in the world[1]. However, economic performance varies considerably among African states due to resources, conflict and governance. In Kenya, where the research was carried out, about half of the population lives in extreme poverty, surviving on less than $2 US a day currently not adjusted to Purchasing Power Parity (PPP). Even though there has been a steady improvement in health outcomes, the country still records incidences of infectious diseases due to poor leadership, corruption, inadequate numbers of health care workers and weak policies in the public health sector [2, 3].

In the mid-20th century, researchers and policy makers began shifting their thinking about health from a biomedical model to a more social determinants of health model. Subsequently, we began to think not just of illness, disease, and death but health and wellbeing. While health is a resource for everyday living and is a positive concept that emphasizes social and personal resources as well as physical capabilities [4] and psychosocial health, wellbeing represents a multidimensional concept with varying operational definitions and interpretations [5]. We adopt Angus Deaton's conceptualization of wellbeing

> as all the things that are good for a person that make for a good life. Wellbeing includes material wellbeing such as income and wealth; physical and psychological wellbeing, represented by health and happiness; education and the ability to participate in civil society through democracy and the rule of law [6].

The World Health Organization (WHO) reports that Neglected Tropical Diseases (NTDs) compromise the health and wellbeing of populations [7, 8]. The term "NTD" was coined during a stakeholders meeting held in 2003 and 2005 in Berlin, Germany, to set the pace for global NTD initiatives [9]. Currently, there is no standard definition of the term "NTD," but several

groups attempt to describe its features. For example, the Public Library of Sciences (PLS) defines NTDs as a group of chronic infectious diseases that promote poverty and are majorly found in Low to Middle-Income Countries (LIMC). According to the PLS definition, NTDs advance poverty due to their negative impact on child health and development, worker productivity and stigmatizing features [10, 11].

Similarly, the Centers for Disease Control and Prevention (CDC) define NTDs as a group of parasitic, bacterial and viral diseases that cause illness and disability to more than 1 billion people worldwide [12]. Based on the CDC definition, Liese et al. [13] propose that NTDs be defined based on disease characteristics, shared features and their effects on poverty and development. For this paper, we adopt Geary [7] and Gyapong and Boatin [8] definition of NTDs that categorizes them as diverse groups of communicable diseases or conditions that globally infect or affect more than 2.7 billion of the most impoverished populations living in Low to Middle-Income Countries (LIMC) of Africa, Asia, and Latin America [14]. Sub-Saharan Africa makes up 90% [10] of the disease burden due to widespread poverty and the distinct characteristics of some NTDs to thrive in specific climates [8]. In Kenya, 18 out of the 20 NTDs listed by the WHO are suspected, confirmed, or endemic to the country [15].

In general, NTDs cost developing economies billions of dollars every year in lost revenue by interfering with labour productivity (agricultural and industrial) and with the wage-earning potential of individuals who are already poor and surviving on less than the US $ 2 a day [16]. Similarly, NTDs limit educational opportunities for school-aged children by interfering with cognitive development and causing undesirable effects on school attendance and child development. Moreover, NTDs trap individuals in a cycle of poverty, leading to social stigma at the family and community levels [17].

Most NTDs are profoundly disabling and account for more than 57 million Disability Adjusted Life Years (DALYs) [14, 18]. Even though DALYs are widely used to estimate the lasting impact of NTDs, they do not measure the full political and socioeconomic implications of the diseases that often maim rather than kill [17]. Hence, NTDs remain endemic to many regions left behind by socioeconomic progress and cause a heavy burden of disability that exceeds malaria and tuberculosis [19]. Furthermore, the long-term nature of NTD infection is complicated by the lack of healthcare access in many areas [20].

In sub-Saharan Africa, political and economic power manifest at different scales, for example, the national, the household and the individual, which determine human-environmental interaction [21], distribution of resources, and the transmission of infectious agents. A political ecology of health approach provides an avenue to understand disease transmission [21], since health and disease are socially constructed in place, this requires an understanding of the broader determinants of health [22, 23]. Historically, biomedicine and the distribution of services through the formal health care system was the sole channel through which the study of disease, health, and healthcare was possible. However, in recent times, health research has acknowledged the importance of socio-environmental factors in the transmission of disease both at the individual and community levels [23, 24]. Thus, this paper uses political ecology of health to examine the political, economic, social, and cultural factors that propagate NTDs in post-devolution Kenya and their effects on health and wellbeing in Kenya.

## Theoretical framework

Human health is understood to be at the interface of social and ecological systems that intersect across spatial and temporal scales. This paper applies the political ecology of health framework to systematically demonstrate how political, economic, and social-cultural systems [25] influence human behaviour and the transmission of infectious diseases [26, 27]. Thus, the

political ecology of health theory situated within the sub-discipline of health geography attempts to connect the large-scale political, economic, social, and cultural processes that shape health and wellbeing.

Stiglitz [28] reports that politics determine how government functions are carried out; when power is unevenly distributed; it creates a loophole for the exploitation of the poor and preserves inequalities. Unequal power yields high poverty rates, illiteracy, health, gender inequality, conflict, and war [6]. Comparatively, conflict disrupts government structures leaving citizens vulnerable and prone to bureaucracy with less effective governments and poor accountability [29]. Moreover, conflict or warfare increases the prevalence of NTD infection because the resources assigned for public health programs are diverted to offset warfare costs. In sub-Saharan Africa, three out of six countries with the highest NTD prevalence have an ongoing violent conflict [29]. This holds for the Central African Republic (CAR), endemic for Human African Trypanosomiasis (HAT), South Sudan for visceral leishmaniasis and leprosy, and Ethiopia for trachoma. When this occurs, politics, which is often seen as a means of improving the social and emotional wellbeing of citizens by advancing their economic circumstances, fails to achieve this objective [30].

Poverty enhances NTD infection, and infection with NTD, in turn, leads back to poverty. Many communities affected by NTDs are of low socioeconomic status with minimal access to health services. In addition to the physical and psychological discomfort caused by NTDs, these diseases pose a tremendous economic burden to individuals, households, communities, and societies. Previous studies conducted in SSA demonstrate that the economic impacts of NTDs such as Lymphatic Filariasis (LF) are likely to be underestimated due to stigma. In instances where individuals abandon work [31], agricultural productivity is affected, which decreases family income [32], leading to malnutrition. Other conditions caused by NTDs such as dengue hemorrhagic fevers and clinical rabies require intensive care, which is quite expensive. Even though the progression of rabies can be prevented through rapid immunization after exposure, the vaccine remains costly and unavailable in many countries of sub-Saharan Africa [9].

The social stigma attached to NTDs, particularly the highly disfiguring ones such as Buruli ulcer, leprosy, lymphatic filariasis, and onchocerciasis, is devastating. Stigma encourages exclusion, rejection, blame, and social judgment [33]. Additionally, stigma leads to social exclusion, reduced quality of life, and poor mental health. As such, there is a need to address the social underpinnings of NTDs in the SDG era to set the pace for shared prosperity, 'leaving no one behind' [19, 34].

Economic activities such as fishing, crop farming, nomadism, and livestock rearing may predispose communities to NTDs such as schistosomiasis, Human African Trypanosomiasis (HAT), and leishmaniasis, respectively, due to the environments in which these livelihoods are pursued [35]. Most of the economic activities in Kenya are influenced by culture and lifestyle factors. For example, pastoralism or nomadism, which is the movement of people from one area to another in search of water and pasture for their animals, may lead to new exposure to disease-causing agents [36].

Culture influences human activities and the community's perception of the interaction between disease and the environment; also, it determines social behaviour and response [35]. As such, it is not unusual, especially in African countries, to find NTD infections being attributed to 'charms' or 'witchcraft.' For example, in Benin, infection with Buruli ulcer is attributed to trespassing on another person's property [37]. Similarly, infection with lymphatic filariasis in Northern Ghana is attributed to witchcraft [38], which leads to stigma. Despite the cultural or religious diversity across many African countries, the experience of stigma seems to be similar in several settings. Stigma confines a source of danger, be it a society's culture, identity or norms, and its mechanism of action is exclusion [39]. As such, stigma affects mobility,

relationships, marriage, employment, and participation in leisure activities, religious and social functions.

Over the last two decades, alterations in the physical environment through agriculture, forestry, industrialization and urbanization have influenced the human use of the environment, exposure to vectors and vulnerability to infection [27]. Also, political instability and limited resources have affected how governments manage environments, control disease transmission, and ensure proper health delivery [27]. Thus, it is vital to explore how political, social-cultural, and economic systems shape health and wellbeing [23].

A few scholars use political ecology of health in the study of infectious diseases; for example, King [25] uses political ecology of health to examine multiple health discourses and environmental systems on the delay of anti-retroviral (ARV) policies in post-apartheid South Africa during the HIV/AIDs pandemic. King [25] also uses political ecology of health to understand the underlying structural factors that propelled the cholera outbreak in Zimbabwe. Hence, political ecology of health provides an understanding of the dynamics of disease transmission and how governments and health care providers respond to the health needs of its population [40, 41] in light of competing political preoccupations and decisions. Political ecology of health examines ecological and socio-political factors, which are important when addressing issues around wellbeing, sustainability, and equity. For example, scholars like Richmond et al. [42] use political ecology of health to capture how the Namgis (Aboriginal people of Canada) think about themselves, relate to each other and their environment. The study establishes that the Namgis define their health and wellbeing in terms of political, economic, social, and cultural aspects of their lives and the ease with which they access natural resources.

## Methods

### Study context

**Ethics statement.** The study protocol was reviewed and approved by the University of Waterloo Research Ethics Committee (ORE#22493) and Maseno University Ethical Review Board (MSU/DRP/MUERC/00496/17). Formal consent was obtained in written form from the participants.

Kenya is in the eastern part of Africa and has a total area of 582,650km$^{2.}$ The population is estimated to be 47,564,296 million people [43]. The country's Gross Domestic Product (GDP) in the year 2017 was US $ 78.76B, and currently, the country ranks 125th out of 157 in the achievement of the Sustainable Development Goals (SDGs) [44]. Kenya has a tropical climate, with 80% of the land consisting of arid and semi-arid zones. Even though 20% of the land is arable, agriculture remains the main economic activity and practices such as irrigation and fishing are considered risk factors for the spread of NTDs such as schistosomiasis [15].

In the year 2010, the promulgation of the new constitution devolved Kenya into 47 administrative counties. Under this system of governance, most of the health functions, including NTD programs, were devolved to the county level. However, the national NTD unit was made responsible for policy formulation, capacity building, monitoring and evaluation of NTD activities in the counties [15]. The Ministry of Health first launched the National multi-year strategic plan of action for the control of NTDs (2011–2015) in November 2011. The conception of this roadmap saw promising steps towards managing NTDs through the WHO recommended strategies for prevention and control. The current (2016–2020) strategic plan aims to scale up access to interventions and treatment of populations at risk; enhance monitoring & evaluation activities; supervise surveillance and operational research activities; strengthen government ownership, coordination and partnership in line with the renewed global momentum towards elimination and eradication of (some) NTDs by the close of 2020 [15].

The research was carried out in the NTD endemic counties of Kenya and explored the meaning and experiences of how NTDs affect the health and wellbeing of communities in Kenya. The research used qualitative methods to provide a deeper understanding of the values, beliefs, and attitudes of participants in the context of NTDs [45]. It utilized country level NTD data to determine areas of NTD prevalence and interventions as per the 2016–2020 NTD strategic plan [15]. The use of qualitative methods [45] allowed for a better understanding of the discourses around inequalities. Data collection

The study used purposive sampling to recruit participants from five NTD endemic counties (Turkana, Kilifi, Busia, Kisumu, and Nairobi). All selected counties were endemic for more than one NTD, and as such, the study enrolled participants who were infected or affected by NTDs aged 18 years and above; both male and female and excluded persons who were below 18 years of age, not infected/affected by NTDs and residing outside the study areas. Data collection took place from December 2017 to February 2018.

A total of 21 Key Informant Interviews (KIIs) were conducted with participants from the national level, including NTD managers, policymakers and with participants from the community level, including community health volunteers, village elders, and local NTD partners (Table 1). An interview guide listing all the topics and issues of interest for the interviews was used, and subtle probes were applied to get detailed information. Interviews were carried out in either English/Swahili and lasted between 30 and 90 minutes. Field notes were taken to supplement the audio recordings, and all voice data was transcribed verbatim. Pseudonyms were used to ensure the participants' privacy.

A total of five Focus Group Discussion (FGDs) of 7–8 participants each (Table 2) were undertaken, ranging in length from 45–90 minutes. Most of the focus group discussions were conducted in the national language of Swahili because the study populations were heterogeneous. A structured FGD guide provided a general overview of the topics during the discussions. All the voice data from the FGDs was captured on audio recorders and transcribed verbatim.

## Socio-demographics of the study participants

The digital recordings of interviews and focus groups were transcribed verbatim for subsequent thematic analysis. Deductive codes captured themes that corresponded to the interview

**Table 1. Characteristics of key informants.**

| Name | Number recruited (total 21) |
|---|---|
| **Role in the community** | |
| Policymaker | 12 |
| Community leader | 3 |
| Researcher | 3 |
| Non-Governmental organization (NGO) | 3 |
| **Gender** | |
| Male | 16 |
| Female | 5 |
| **Age** | |
| 18–35 years | 6 |
| 35–60 years | 13 |
| Over 61 years | 2 |
| **Years of work** | |
| 1–5 years | 15 |
| 5–10 years | 6 |

**Table 2. A table showing the demography of the focus group participants.**

| Name | Number recruited (total 46) |
|---|---|
| **Place of birth** | |
| Born in the community | 32 |
| Born outside the community | 14 |
| **Length of stay in the community** | |
| Less than 5 years | 10 |
| 5–10 years | 12 |
| More than 10 years | 24 |
| **Economic activities** | |
| Casual laborers | 6 |
| Salaried workers in the county | 2 |
| Small scale farmers | 11 |
| Small business operators | 13 |
| Unemployed | 14 |
| **Gender** | |
| Female | 21 |
| Male | 25 |

questions, existing literature, and concepts, while inductive codes considered issues emerging from the transcripts. Two transcripts were independently coded for each data source to establish the inter-rater reliability. The aim was to determine at least a 70% agreement for each source, as described by Miles and Huberman [46]. The agreed upon thematic codes were subsequently uploaded on NVivo version 12 and applied to the transcripts.

# Results

## Political factors

Whereas the clustering of NTDs has made it relatively easier for intervention strategies to be carried out, we find there are barriers to effective preventive, diagnostic, and curative services for NTDs in the form of political, economic, and social-cultural interests (Table 3).

The frequently mentioned political factors among the key informants include lack of political will from the county government, inadequate budgetary allocations to address and implement NTD control strategies at the county level, and frequent health worker strikes (Table 3).

**Table 3. Political factors affecting NTDs in Kenya.**

| Political Factors | Key informants (KIIs) (n = 21) | | Focus groups (n = 5) Total participants (46) | |
|---|---|---|---|---|
| | # of Key informants | # of mentions | # of FGDs | # of mentions |
| • Devolution of government services | | | | |
| • Lack of political will | 8 | 27 | 2 | 9 |
| • Poor budgetary allocations | 7 | 40 | 1 | 4 |
| • Health workers strike | 6 | 16 | 5 | 16 |
| • 'Political appointments' | 2 | 7 | 0 | 0 |
| • Denial of NTDs | 2 | 5 | 1 | 6 |
| • Other initiatives vs. NTDs | 4 | 6 | 1 | 3 |
| • Political tensions, violence and skirmishes, Border tensions and conflicts | 3 | 7 | 3 | 5 |
| • 'Politicize' essential resources | 1 | 2 | 3 | 3 |

On the other hand, the FGDs participants raised the issue that devolution is a means by which residents have been able to access jobs. However, as mentioned by the key informants, the community members felt that the county governments had frustrated health workers leading to frequent strikes that interrupt the provision of medical services in the hospitals. This is because devolution is a new and ongoing concept that involves the decentralization of government services to improve accessibility and job creation:

*'First of all, I would say that the devolution is still ongoing; the process is not yet complete. A number of times the nurses have been on strike, doctors have been on strike the clinical officers, the technicians, and whenever there are strikes, enrollment just crashes. No one is coming to the hospital'* (Mary, female 30 years, **KI**).

Key informants affirm that devolution led to the premature transfer of health services from the national government to the county government, leaving all health decisions to be made by 'political appointees' (Table 3) without appropriate training. This factor was not mentioned by any of the focus group participants, but this key informant had this to say:

*'You find that when something comes from the assembly, a technical person may give the direction but with them [political appointees]they will come up with the final decision which suits them, so that is why you will find that most of the NTDs are not addressed because you know politicians are lay people they want to see something come out as an outbreak that is when they know it is a problem'* (Peter, male 34 years, **KI**).

This kind of arrangement creates an avenue for NTDs to spread, coupled with inadequate budgetary allocations:

*'Most of the times, there are no funds allocated just as the name suggests, these are neglected tropical diseases. So with the politician, if you tell them that this is a neglected disease, they will tell you, do you want me to initiate the program if the Ministry itself has called it neglected*?' (John, male 32 years, **KI**).

Both the FGDs participants and the KIs mentioned that some political leaders deny (Table 3) the presence of NTDs in their area because they do not want their areas to be associated with diseases of poverty, which creates stumbling blocks for research and interventions: *'Most politicians would not want their areas to be known to have jiggers because more or less, you are exposing them that they are not doing much'* (John, male 34 years, **FGD**).

Other political barriers include politicians preferring to implement specific initiatives (Table 3) over pressing NTD interventions, for example, constructing health facilities as opposed to performing facial cleanliness and environmental sanitation (F&E):

*'So there is the prioritization of resources at the county level, most politicians would want to support programs or projects like the physical constructing of health dispensaries, and during the next campaign, they will say you see I am behind this project. So they despise non-tangible issues like F& E' [facial cleanliness and environmental sanitation]* (Kanji, male 24 years, **KI**).

Also, politicians tend to 'politicize' essential resources, for example, water in dry habitats. In such contexts, water is used as a weapon to entice or punish voters, which leads to low political engagement, marginalization, and the spread of infectious diseases:

*'You know, water here is a precious commodity. And most NGOs [Non-Governmental Organizations] when they came in, they dug shallow wells, they assigned the shallow wells to certain individuals. The community itself appointed the chairman, the treasurer, and the secretary of the water users association. They formed the water users association so that they can take care of this water [water politics]. So you find that this water point is accruing a lot of money at the same time it is also earning the interest of the politician. So during campaigns, it is used as a tool to punish those who do not vote for so and so'* (Pamela, female 44 years, **FGD**).

In counties that border neighbouring countries, for example, Turkana, tension, conflict, and insecurity are quite common, and this hampers NTD control efforts. *'We migrate as far as Ethiopia due to tension and conflict; we become a population that cannot access a health facility, and we really suffer from many diseases, especially Kalazar [NTD]'* (Kamara, male 40 years, **FGD**).

## Economic factors

The focus group respondents engage in various economic activities to earn a living, including running small businesses such as fish and milk vending, palm wine tapping, charcoal burning, and motorbike operation. Others practise small scale farming and pastoralism, casual labour in irrigation and fishing sectors, salaried work at the county governments, and the rest are unemployed (Table 2). The economic impacts of NTDs are diverse, and both the Key informants and the FG discussants mentioned that NTDs affect economic productivity and capacity to earn an income leaving infected persons with little or no money to access health, education and other opportunities (Table 4).

*'This disease affects us because of lack of money. When you are sick and do not have money at all, you can't take care of yourself at all. Hospitals are there, but you cannot go to the hospital, so you are affected. Your economic activity becomes limited, and your earnings also decrease, and you cannot work at all. That is what affects us. In this economy, if you do not have the money, you cannot support yourself and you cannot have an earning because you are sick'* (Halima, female 29 years, **FGD**).

**Table 4. A table showing the economic, social and cultural factors of NTD infection in Kenya.**

| Economic Factors | Key informants (n = 21) | | Focus groups (n = 5) Total participants (46) | |
|---|---|---|---|---|
| | # of Key informants | # of mentions | # of FGDs | # of mentions |
| • Affect economic productivity | 21 | 57 | 5 | 80 |
| • Lack of economic opportunity | 7 | 15 | 5 | 31 |
| • Little or no income | 4 | 42 | 5 | 74 |
| **Social Factors** | | | | |
| • Low levels of literacy | 20 | 72 | 5 | 43 |
| • Marginalization in social affairs | 7 | 60 | 4 | 5 |
| • School absenteeism and dropouts | 7 | 57 | 3 | 8 |
| • Poor advocacy | 6 | 26 | 2 | 15 |
| • Desertion | 5 | 17 | 4 | 21 |
| • Domestic and marital conflict | 3 | 14 | 5 | 12 |
| **Cultural Factors** | | | | |
| • Cultural norms and practices | 7 | 29 | 3 | 1 |
| • Curses | 2 | 5 | 2 | 4 |

A key informant respondent observed the economic impact of NTDs at a national level:

*'NTDs interfere with the people's output, so at the individual level, there is a huge impact on the economic productivity at a community level; it is even bigger. It is a much bigger impact; we are estimating that annually we are losing about 200 billion Kenyan shillings [1,872,951,200USD] every year to NTDs, and that is a very conservative estimate as a country. Of course, there is a big impact on productivity, they also have a big impact in terms of bills incurred in treatment, so generally, there is a big dent that the NTDs are having on the economic situation. Not just at an individual but also at a community level'* (Simon, male 40 years. **KI**).

Individuals suffering from disabling NTDs such as leprosy are heavily discriminated against because of their wounds and disfigured limbs; if they are involved in any form of economic opportunity, it is bound to be unsuccessful because prospective customers perceive their physical state as infectious:

*'You know even if she wanted to sit in a 'kibanda'- [in a kiosk] or maybe be a fishmonger, people would not buy the fish because of the condition of the fingers, even the ones with the jiggers. When one is down with it, they are not able to be productive in the community'* (Simon, male 40 years, **KI**).

## Social and cultural factors

Both the KI and the FG discussants agree that NTDs thrive in areas which have low levels of literacy which leads to ignorance, for example, in the case of transmission of Chikungunya:

*'The disease, I heard that it was in Rwanda from the news. It was in Rwanda, then it was in Mombasa, but I was not aware of how this disease is, till I got infected. The other day, I told people that I have Chikungunya, and when I went to class, everybody was like, do not give us Chikungunya. So I thought it was something like when you sit with someone you infect them'* (Janet, female 25 years, **FGD**).

The engagement in social affairs was explored in terms of family and community support, participation in social gatherings such as Baraza's (chiefs' meetings), church activities, volunteer activities, 'merry go rounds' and women groups. The results indicate that people infected with NTDs are unable to take part in communal events due to shame, and information distortion, which makes them marginalized in social affairs (Table 4). *'Those infested with jiggers are peasant farmers, and they are poverty-stricken families or communities, and are more or less neglected in social affairs, the current social affairs'* (Stella, female 27 years, **KI**).

Suppose the infected individuals have information on the communal gathering at hand, the physical challenges associated with the disease makes it difficult for them to attend and participate: *'Am not able to participate socially because my legs are swollen'* (Amina, female 54 years, **FGD**).

Even if they attend communal gatherings, they are not able to concentrate. *'When you participate in social gatherings, once the jigger starts to bite or itch while you are in the social gathering, you forget about the social gathering'* (Joyce, female 28 years, **FGD**).

In children, most of the KI mentioned that NTD infection is a promoter of school absenteeism and dropouts and a consequence as seen in the increasing illiteracy rates (Table 4) as mentioned by the respondent below:

*'I know what we look at greatly is school attendance. Two, you might have attendance, but you do not have the other bit of school attendance, which is the concentration in class. Furthermore, the third would be cognitive development and being able to learn. I think that because of malnutrition and the growth of the worms, it impairs even the mental acuteness of the child, and the child is not able to learn at a fast pace which delays their schooling'* (Caleb, male 36 years, **KI)**.

Similarly, social challenges associated with NTD infection include desertion, poor advocacy from the affected groups due to stigmatization and marginalization, domestic and marital conflict leading to divorce, abandonment, and collapse of the family structure:

*'These diseases have a lot of challenges because when they have not infected you, you will have friends. However, the moment you get the disease, the friends will be few. This is because any time you are with them, you will be borrowing money to go to the hospital because you are now poor. Secondly, if you do not have the disease, and you have a wife, and you are not unfaithful to her, then your wife has no stress. However, the moment you get sick, even your wife will start thinking you are a burden because you cannot work, and she will initiate other domestic problems in the household [sexual relations], so she goes away'* (Peter, male 27 years, **FGD**).

The cultural factors that influence NTD transmission include norms and practices (Table 4). For example, among the nomadic communities living in arid and semi-arid areas, there are norms prohibiting the consumption of animal proteins such as fish, eggs, and chicken in women and children, leading to protein deficiency and malnutrition in populations that lack access to alternative sources of plant protein. Malnutrition compromises the body's immunity to fight diseases and makes it hard to carry out surgical interventions for NTDs such as visceral leishmaniasis (Kalazar).

*'If the patient is anemic, it may take 2 days for their iron levels to pick up through blood transfusion. If the patients are malnourished, it takes more time for them to be fed so they can recover. Otherwise, if you have kalazar and all your health vitals are optimum, surgery and treatment are immediate, we do not wait'* (Walter, male 52 years, **KI**).

Some communities consider NTDs such as jiggers and leprosy, to be hereditary. *'My mother had leprosy, I have leprosy, so it is like I inherited it. My uncle also had the same case of leprosy'* (Sara, female 49 years, **FGD**).

An NTD such as leprosy is regarded as a curse (Table 4) in some communities: *'It is believed that leprosy is a curse, so people delay seeking medical intervention; they ask, why do we go to the hospital, and it is a curse?'* (James, male 28 years, **FGD**).

## Water, sanitation, and infrastructure

The sources of water in the study areas ranged from boreholes, lakes, dams, rivers, streams, earth pans, springs, rainwater, piped water and 'leak it in' or 'tippy-tap' (a container filled with water and suspended on a tree. The container is perforated at the bottom using a sharp object to expose a hole in which water is released for use when required) (Table 5). Both KI and FGDs respondents agreed that natural water bodies are the primary water sources in the study counties.

*'Water, water, water is a problem. We depend on natural water bodies, and this area is arid, so water is an issue. The Ministry of water is trying to come up with boreholes so that we can*

**Table 5. A table showing Water, sanitation and housing factors in NTD infection in Kenya.**

| | Key informants (n = 21) | | Focus groups (n = 5) Total participants (46) | |
| --- | --- | --- | --- | --- |
| | # of Key informants | # of mentions | # of FGDs | # of mentions |
| **Water sources** | | | | |
| • Boreholes | 7 | 16 | 2 | 12 |
| • Rivers | 6 | 7 | 5 | 28 |
| • Lakes | 4 | 10 | 2 | 2 |
| • Dams | 4 | 4 | 4 | 7 |
| • 'Leak it in' or 'tippy-tap.' | 4 | 1 | 3 | 17 |
| • Earth pans | 3 | 2 | 3 | 21 |
| • Rainwater | 1 | 5 | 1 | 4 |
| • Piped water | 1 | 6 | 2 | 8 |
| **Sanitation facilities** | | | | |
| • Open defecation | 7 | 21 | 5 | 23 |
| • Latrines | 5 | 17 | 5 | 12 |
| • Toilets | 2 | 12 | 4 | 18 |
| **Challenges** | | | | |
| • Open defecation | 3 | 29 | 5 | 40 |
| • Terrain and soil structure | 2 | 2 | 1 | 1 |
| • Poor refuse collection and management | 1 | 3 | 1 | 2 |
| **Housing status** | | | | |
| • Semi-permanent structures | 5 | 12 | 5 | 11 |
| • Indecent | 4 | 17 | 5 | 29 |
| • Poor lighting and ventilation. | 2 | 5 | 5 | 8 |

have a constant supply, but water is an issue. That is why there are nomads. They travel long distances looking for water and pasture' *(*Angeline, female 52 years, **KI)**.

*'The kids wash in the rivers, they wash in the lakes, they fetch water, and they do their chores at the lake barefoot. They also carry their younger brothers and sisters, they go swimming in the lake, they bathe in the lake, and all those are predisposing factors to schistosomiasis'* (Washington male 60 years **KI**).

The few residents, who have access to piped water, have questionable and irregular supply due to water rationing: *'A few areas have tapped water, but it is not evenly distributed there is a bit of rationing'* (Fatuma, female 30 years, **FGD**).

Sanitation facilities are a challenge in most parts of the country, with a number of people practicing open defecation (Table 5) as highly mentioned by the KI and FGDs. *'Here in western Kenya, we border Lake Victoria, but because of poor latrine coverage, we have very many infections of schistosomiasis and STHs. People are still defecating in the open'* (Jacklyn, female 40 years, **FGD**).

Even though several individuals and groups have constructed latrines and toilets, open defecation (Table 5) remains a common practice and a contributing factor for schistosomiasis, STHs, and diarrheal infections. Part of the reasons for the practice of open defecation among the KI and FGDs are terrain and soil structure. Places which have loose, or black cotton soil have a frequent collapse of latrines:

*'Western Kenya, you will find nearly 60% of the area is predominantly black cotton soil, and black cotton soil is quite unstable, so you want to encourage people to dig pit latrines, but then they face a challenge because this type of soil is very loose, so during wet season if you do not do the proper lining, it tends to collapse. Alternatively, even during the dry season, when you do not line that pit, then it tends to crack, and therefore you find that you put up a latrine and every other season it will collapse'* (Mark, male 50 years, **KI**).

Additionally, there is poor refuse collection and management in urban and peri-urban areas, which provides an avenue for dengue transmission.

*'Waste disposal in this community is about you and your family. If you have generated waste in the family, you have to find a way to burn them, because we do not have those vehicles that come to collect waste'* (Janet, female 43 years, **KI**).

## Housing status

Both the KI and FGDs mentioned that housing facilities in communities affected by NTDs are indecent due to poverty. The households have dusty floors consisting of top and loose soil which harbours vectors such as fleas and mosquitoes:

*'Housing plays a major role in the transmission. These poor families they cannot afford decent housing or even the way they construct the houses are poor. When I was young, I used to see when someone was constructing a house or 'simba' [hut] they used to dig the top vegetable soil and throw it away. That top vegetable soil is ideal ground for a jigger to manifest or multiply in. So nowadays I do see most of the houses that are being constructed they just dig the hole for the posts, they put it in, but they do not remove the top vegetable soil they just use it as a floor'* (John, male 25 years, **KI**).

Similarly, another respondent from the FGDs added:

*'We have loose soil in our house, and this alone contributes to the spread of the Jiggers infestation since the fleas are found there, and children are running barefooted in the house if the neighbour's children come to play, they share in the jigger infestation'* (Jack, male, 25 years, **FGD**).

*'Poor housing promotes NTDs, the dome shaped structures are dark, and mosquitoes like hiding in dome-shaped structures and that is how you end up with vector-borne NTDs'* (Brian, male 43 years, **KI**).

Furthermore, in areas like the coastal region, housing status contributes to NTD transmission as noted:

*'There is a myth in the place that I work that "NTDs; for example, hydrocele is inherited". So in relation to housing structure, our houses are built in such a way that they do not have ventilation. And they are cladded with grass, so this grass harbours the mosquito. So if this mosquito being the vector carrying the parasite, it can stay in that house for many years. So it is only transferring the parasite from this person to the other person, so for the community, they say that this is a disease that has been inherited, not knowing that it is the housing structure that harbours this mosquito who carries the parasite to the other person. So surely, the housing structure contributes to some of the NTDs'* (Kanji, male 45 years, **KI**).

## Discussion

Our results show that broad structural factors propagate inequity and NTDs in the context of poverty, which affects population health and wellbeing. We find that people infected and affected by NTDs are unable to lobby for fundamental human rights, possess limited finances, lack access to health care and live in areas that have polluted water sources, poor sanitation, and housing.

The devolution process in Kenya, which aimed to strengthen democracy and accountability, increase community participation, improve efficiency in the delivery of public systems, and reduce inequities [47] came with various challenges. For example, the health ministry which is the largest devolved sector is still experiencing human resource deficiency, corruption, political interference, and infrastructural shortcomings [48] almost ten years later.

We find that prioritizing political initiatives over NTD control programs propagates NTD infection in Kenya even though it is suggested that political action may be a useful channel of controlling NTDs, through poverty elimination, social cohesion, and economic development [49, 50]. We further establish that obstacles in the formulation of NTD policies, for example, health service management at the county level, direct the health care provision and management of NTD activities; however, a health workers' strike affects the diagnosis and treatment of NTDs. Other political challenges, such as goodwill, "political appointments," and inadequate budgetary allocations, frustrate and interfere with NTD programs. In line with our observations, Jaumotte et al. [51] argue that powerful economic interests, inequitable distribution of social resources, and tension between national and local authorities [49, 50] may delay NTD activities and hinder societal wellbeing [51].

Devolution has changed the locus of power from central to sub-national levels setting the pace for priorities at the local level due to limited resources. As a result, the current public reforms are being driven by the contextual desire for a positive outcome, especially in counties which were previously neglected. Also, rapid uptake of the devolution combined with limited technical and decision-making capacity, the public delivery of services has been distorted for political interests [47]. For this reason, we find that poor political accountability, inconsistent budgetary allocations, and rampant health workers' strike in the context of devolution has driven the poverty-NTD cycle. Compared to other countries in sub-Saharan Africa, Kenya has managed to successfully implement large scale NTD programs such as schistosomiasis and STHs control using the School-Based Deworming program (SBD). However, it is maintained that the success of such programs is dependent on successful collaborative partnerships, political goodwill, accountability, and strong Monitoring and Evaluation (M & E) frameworks, which ensures that the stated objectives are achieved [52].

Our findings on political interference in the delivery of public health care are consistent with the observations from Ottersen et al. [53], who recognize that the delivery of essential health care services may sometimes fall on non-state actors, in our case, the delivery of essential health services fell on "political appointees" who were not technical people but had the capacity to use their power and influence to advance NTDs. We also confirm that power and politics do not only affect governance but remain critical factors in the delivery of public health care. In terms of health care accountability, our results indicate that politicians prefer to implement specific "tangible" initiatives over pressing NTD interventions, for example, constructing non-essential buildings over the promotion of environmental sanitation. Comparably, McCollum et al. [47] recognize that with devolution politics comes less visible community services that focus on health promotion, disease prevention and referrals to more of tangible curative health services. Even though our results affirm that international partners are key in the delivery of local NTD programs, we also concur that governments in NTD endemic regions must make a political commitment towards NTDs if the greatest impact is to be achieved [54].

The implementation of NTD programs occurs in varied social-economic contexts. As such, our results suggest that NTDs may affect the economic opportunity and productivity of infected persons. We find that persons affected by NTDs are unable to work and earn an income on a local scale, which affects the national economic output. In addition to NTDs impacting the general health and life expectancy of impoverished populations, its economic consequences include the reduced ability to work. Comparably, Lenk et al. [55] find that the economic impact of NTDs depends on the type and severity of infection as well as the context in which the disease occurs. In our case, we did not quantify the impacts of the specific NTDs, but our results consistently demonstrate that NTDs affect economic productivity at the individual and national level.

Our analysis confirms that NTD infection contributes to school absenteeism and dropouts in school age children, which later manifest as low levels of literacy in adulthood, and consequentially poor advocacy and marginalization in social affairs. Comparatively, Norris et al. [31] document that NTDs reduce school attendance, results in stunted growth, impairs cognitive development in children, and promotes the social isolation of the already stigmatized populations. Moreover, NTDs are a consistent cause of domestic and marital conflict as well as desertion practices, which disproportionately affects women. The scarring and disfigurement resulting from NTD infection may prevent young women from getting married or act as grounds for spousal abandonment. Our results are consistent with the observations from Hunt et al. [56], who establish that NTDs inflict a substantial economic burden to individuals, households, communities, and society through a loss in economic gain, increased inpatient stays, high rates of school abseeitism and gender inequality all of which promote poverty and ill health for populations.

Cultural practices play a role in NTD transmission; for example, prohibiting the consumption of highly nutritious foods compromises the body's immunity to fight diseases and makes it hard for surgical interventions in NTDs like visceral leishmaniasis due to malnutrition. In light of this, Hall et al. [57] affirm the direct relationship between undernutrition and NTDs. The authors argue that a lack of proper nutrition increases the risk of infection, the severity of disease, impairs host response to the pathogen and may be a cause of death. In our research context, we found that cultural practices determine the community's perception of the cause of NTD transmission. For example, most of our study participants attributed infections with leprosy and jiggers to curses and hereditary transmission, respectively. Similarly, previous studies by Ahorlu et al. [38] report that negative cultural perceptions on the causes of disabling NTDs, such as lymphatic filariasis infection, hamper health seeking behaviour in infected persons. As such, our research establishes that the political, economic, social, and cultural factors enhance inequities that propagate NTD infection in Kenya. Correspondingly, Aagaard-Hansen and Chaignat [58] suggest that for health care systems to be responsive to the needs of all people, they must be sensitive to the local political, socio-cultural, and environmental structures.

Water remains a limited commodity in most NTD endemic regions of Kenya. We find that in certain situations, influential political figures use water scarcity as a weapon to "punish" community members who disagree with their political ideologies. An overreliance on natural water bodies such as rivers, lakes, and streams for daily use contributes to NTD infection since natural water sources are contaminated with vectors. We find that piped water is expensive to connect and maintain, and in areas where it is available, there are incidences of water rationing and questionable water quality. Similarly, sanitation facilities are wanting in both urban and rural areas with problems such as open defecation becoming risk factors for schistosomiasis and STHs. Other infrastructural problems that propagate NTD infection in Kenya lies within the housing sector with the current housing facilities in NTD endemic regions consisting of

indecent, semi-permanent structures with poor lighting and ventilation that harbour parasitic vectors.

Our research makes substantial contributions to knowledge, policy, and practice. We use social theory to demonstrate that political, economic, social, and cultural factors propagate NTD infections in Kenya. Furthermore, we use qualitative methods to highlight the lived-in experiences of persons infected and affected by NTDs in the largely biomedical field of infectious diseases. Despite our research contributions, our research had some limitations as well. First, our research was qualitative and was based on self-reported data, which is susceptible to small sample sizes and social desirability. In our case, our participants could have overstated or understated their experiences, depending on expectations. To minimize research bias, we asked our participants (KIs and FGs) follow up questions, and we used subtle probes to generate further explanations on their NTD experience. Second, our study was cross-sectional, which means we collected data at a single time point for the KIs who were interviewed alone and the focus group discussants who were interviewed in a group; as such, our study design did not allow for the examination of potential changes over time in light of the ongoing country-wide NTD strategies. However, with the triangulation of methods, we were able to minimize bias and ensure similarity in the emerging themes between the FGDs and the KIIs.

In conclusion, the goal of this paper was to identify the political, economic, social, and cultural factors that propagate NTDs and their effects on health and wellbeing. To accomplish this, we used political ecology of health in the largely biomedical field of NTDs to provide an explanatory power and demonstrate that macro-level factors in society propagate NTD infection. Our research recognizes that place-based experiences shape the health and wellbeing of marginalized populations infected or affected by NTDs. Thus, our research broadens NTD discourses away from biomedical preoccupations towards extensive commitments to the causes of inequity as propagated by the political, economic, social and cultural factors for the formulation of equitable and inclusive policies around empowerment and poverty alleviation. As such, we recommend that post-devolution Kenya ensures that local political, economic and socio-cultural structures are equitable, sensitive and responsive to the needs of all people. We also propose poverty alleviation through capacity building and empowerment as a means of tackling NTDs for sustained economic opportunity and productivity at the local and national level.

## Author Contributions

**Conceptualization:** Elizabeth A. Ochola, Diana M. S. Karanja, Susan J. Elliott.

**Data curation:** Elizabeth A. Ochola, Susan J. Elliott.

**Formal analysis:** Elizabeth A. Ochola, Susan J. Elliott.

**Funding acquisition:** Susan J. Elliott.

**Investigation:** Elizabeth A. Ochola.

**Methodology:** Elizabeth A. Ochola, Diana M. S. Karanja, Susan J. Elliott.

**Project administration:** Diana M. S. Karanja.

**Resources:** Diana M. S. Karanja, Susan J. Elliott.

**Software:** Elizabeth A. Ochola, Susan J. Elliott.

**Supervision:** Diana M. S. Karanja, Susan J. Elliott.

**Validation:** Diana M. S. Karanja, Susan J. Elliott.

**Visualization:** Susan J. Elliott.

**Writing – original draft:** Elizabeth A. Ochola.

**Writing – review & editing:** Elizabeth A. Ochola, Diana M. S. Karanja, Susan J. Elliott.

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
