## [Decision Letter · Decision Letter 0]

15 Aug 2020

Dear Dr. Ochola,

Thank you very much for submitting your manuscript "The Impact of Neglected Tropical Diseases (NTDs) on Health and Wellbeing 

in sub-Saharan Africa (SSA): A Case Study of Kenya" for consideration at PLOS Neglected Tropical Diseases. As with all papers reviewed by the journal, your manuscript was reviewed by members of the editorial board and by several independent reviewers. In light of the reviews (below this email), we would like to invite the resubmission of a significantly-revised version that takes into account the reviewers' comments. 

We cannot make any decision about publication until we have seen the revised manuscript and your response to the reviewers' comments. Your revised manuscript is also likely to be sent to reviewers for further evaluation.

Sincerely,

Victoria J. Brookes

Deputy Editor

Victoria Brookes

Deputy Editor

Reviewer's Responses to Questions

**Key Review Criteria Required for Acceptance?**

**Methods**

-Are the objectives of the study clearly articulated with a clear testable hypothesis stated?

-Is the study design appropriate to address the stated objectives?

-Is the population clearly described and appropriate for the hypothesis being tested?

-Is the sample size sufficient to ensure adequate power to address the hypothesis being tested?

-Were correct statistical analysis used to support conclusions?

-Are there concerns about ethical or regulatory requirements being met?

Reviewer #1: This is a qualitative study. As such, its aim is not to "test" hypotheses but to explore the qualities of a specific phenomenon. The study design is appropriate to explore the aim stated in the Author's summary (although there is a discrepancy in clarity between this and the abstract).

The sample size, as it is often the case with qualitative studies, is small. However, the data produced in the text and discussed by the authors show consistently recurrent themes. 

The thematic analysis, discussed in the methods and exemplified in the tables, is easy and clear to follow and it is coherent with the Discussion section.

I have no concern about ethical of regulatory requirements. The authors state that they have used pseudonyms, and I did not find details that could lead to identification of participants.

Reviewer #2: A very interesting and relevant article which applies the political ecology of health framework to interrogate health discourses as produced in society to demonstrate how political, economic and social-cultural systems influence human behavior and the transmission of infectious diseases.

My key concerns, which i believe if addressed will strengthen the manuscript, are:

1. The authors do not provide a clear definition for NTDs. While this may be an NTD journal, it may be worthwhile to consider that a small proportion of readers may not be knowledgeable about this field. It would be beneficial to save them the extra effort of searching elsewhere to understand what NTDs are. Granted that there may not be an overarching definition, but the authors can describe briefly why this group of mostly parasitic and bacterial diseases are considered "neglected". This can strengthen the importance already stated between lines 88 and 95 (with a few examples).

2. Lines 111 to 148 seem to just jump in from no where. There's a need to describe the theoretical basis for considering these factors as important. I suggest that these paragraphs are best placed under the "Theoretical Framework". 

3. Theoretical framework: This section should focus on describing the components of the political ecology of health and how this applies to the study design. Lines 162-175 may not be really necessary. The authors should consider replacing this with the content of lines 111-148.

Reviewer #3: please see attached document

**Results**

-Does the analysis presented match the analysis plan?

-Are the results clearly and completely presented?

-Are the figures (Tables, Images) of sufficient quality for clarity?

Reviewer #1: The analysis presented partially matches the analysis plan. The aim of the manuscript is not stated clearly. In the abstract, the authors mention social, economic, political, cultural impacts of NTDs but, throughout the text, they focus on the impact of social, economic, political, cultural factors “on” NTDs—which, I suspect, is the overall aim. I thus suggest rephrasing the mention in the abstract.

The figures and tables are clear.

Reviewer #2: Tables 3-5: I see little value in describing the number of key informants/ focus groups per sub-theme. It really adds no value to a qualitative research given that the ficus is to elicit themes, as against quantify the distribution of responses to themes.

There is more value if the authors consider including subthemes (with narratives and supporting quotes) under each of the broad themes. For example under Political Factors (which is a deductive theme from the theoretical framework), I'd love to see the emerging themes (such as Devolution of government services and Lack of political will) described in better details with their supporting quotes where necessary. 

This is the best way to prove the statement by the Authors under methods section (Lines 241-243) that "Deductive codes captured themes that corresponded to the interview questions, existing literature, and concepts, while inductive codes considered issues emerging from the transcripts"

Reviewer #3: (No Response)

**Conclusions**

-Are the conclusions supported by the data presented?

-Are the limitations of analysis clearly described?

-Do the authors discuss how these data can be helpful to advance our understanding of the topic under study?

-Is public health relevance addressed?

Reviewer #1: This manuscript lacks a “Conclusions” section and the authors only add a paragraph at the end of the Discussion. While the authors do state their advocacy for a broader approach to tackling NTDs quite clearly, and the public health relevance is evident, they do not satisfactorily engage with previous works and the limitations of the study are not addressed. For instance, small samples are the norm in qualitative studies, but this should still be acknowledged, as well as the fact that individuals were interviewed (alone or in focus group discussions), which are methods with their own sets of bias.

Reviewer #2: Any study limitations?

Reviewer #3: (No Response)

**Editorial and Data Presentation Modifications?**

Reviewer #1: The manuscript needs to be thoroughly proofread. The style and register are inconsistent, there are some conceptual and literal repetitions (see lines 162-163; 231/33 and 240/41) and isolated expressions that do not match the overall tone (174: “jumping off point”; 550: “punish”). The Vancouver style is used without in-text citations (e.g. “as [37] suggests” instead of “as Smith [37] suggests”), which make the reading experience unnecessarily difficult. The Results sections, as reported, are uneven. I suggest merging 1.6 Social Factors with 1.7 Cultural Factors, and renaming 1.8 along the lines of “Water, Sanitation, and Infrastructure”.

Reviewer #2: (No Response)

Reviewer #3: (No Response)

**Summary and General Comments**

Reviewer #1: Descriptive papers highlighting the complexity of persisting diseases are needed, most especially when addressing marginalised groups (often blamed for “poor practices”), and this paper has the merit of bringing together different ‘bottom-up’ perspectives. However, there are a few discrepancies between what the authors claim they will do and what they actually do. The authors mention but do not elaborate on the analytical framework, which remains in the background without tying the data together. The Discussion follows the different sections of the Results but does not engage with the previous literature on the topic, and one is left wondering what is it that they were supposed to notice but did not. The reader needs the authors to tell them where to look among the data collected and how to look at them. 

By contrast, some sentences are both very direct and very confusing, seemingly establishing unidirectional relationships of cause and consequence between factors. For instance, 111-113: “Politics determine how government functions are carried out when power is unevenly distributed; it creates a loophole for the exploitation of the poor, which hinders an egalitarian economy, society and preserves inequalities”. Surely politics determines but also results from the uneven distribution of power. It is also unclear if the authors are referring to Kenya or making a general point. While I agree with the spirit of the sentence, it is vague and convoluted, and it would be improved by some examples (other countries, other authors – not simply referenced but brought in the discussion) to help spell out what they are referring to exactly. In lines 168-169, it is not clear who the subject is. Who is competing with whom?

What are the implications of this paper for contexts that are not the counties involved? I am not sure that the notion of ‘proxy’, mentioned in the Introduction and at the beginning of the Discussion is the most useful. It suggests that one (poverty) can be replaced by the other (NTDs). I am not certain this is the point the authors want to make and the phrasing echoes positivist methods such as regression analysis. 

The authors mention the necessity to explore the social, political, economic factors that shape health and wellbeing (160-161). I agree, but the topic is not new; it has been explored – in different contexts, in different manners. Yet, the authors do not engage with such literature and sometimes simply mention it. The paragraph between lines 140-148, for instance, could be rephrased to imply that the experience of stigma is similar everywhere, because all systems sanction morality. The role of stigma is to confine a source of danger (to society’s culture, identity, norms etc), thus exclusion is the mechanism through which stigma works. I understand that this is not the focus of this manuscript, but leaving concepts such as stigma and witchcraft un-problematised (especially when other disciplines have abundantly done so) on a manuscript that (I hope) will be widely read by policy/global health/epidemiology experts seems counterproductive to the broader aim of this work—which, if I understood correctly, it to expand our perspective on NTDs. I believe the Introduction and the Discussion should dedicate more space to engage with both existing works and policy implications related to the data the authors present. It will strengthen the conclusion of the paper. 

On that note, I suggest a separate section for the Conclusions to which, as it is, only two sentences are dedicated.

Reviewer #2: I believe this is a very important and interesting study. Once the authors are able to address these few concerns, tha article will be stronger and more appealing to a broader readership.

Reviewer #3: (No Response)

PLOS authors have the option to publish the peer review history of their article (what does this mean?). If published, this will include your full peer review and any attached files.

Reviewer #1: No

Reviewer #2: No

Reviewer #3: No
---

## [Decision Letter · Decision Letter 1]

9 Nov 2020

Dear Dr. Ochola,

Thank you very much for submitting your manuscript "The Impact of Neglected Tropical Diseases (NTDs) on Health and Wellbeing in sub-Saharan Africa (SSA): A Case Study of Kenya" for consideration at PLOS Neglected Tropical Diseases. As with all papers reviewed by the journal, your manuscript was reviewed by members of the editorial board and by several independent reviewers. The reviewers appreciated the attention to an important topic. Based on the reviews, we are likely to accept this manuscript for publication, providing that you modify the manuscript according to the review recommendations. 

Although the number of comments from the reviewers are relatively few, please pay particular attention to the comment from R1:

550-557: this is possibly the whole point of the paper, which shows that a specific phenomenon (devolution) in a specific context (Kenya) has ignored the issue of NDTs contributing to their exacerbation. Its importance, however, seems to be diluted throughout the paper in favour of more general statements about NDTs and structural poverty. The paper would benefit enormously from having this preceded by a clear statement in the Introduction (e.g. "This paper focuses on post-devolution Kenya to show the political economy of policy initiatives and its influence on NDTs prevalence"). One or two sentences would help us frame the whole paper and the information therein, giving us a straight line to follow the data.

Sincerely,

Lisa Dikomitis

Associate Editor

Victoria Brookes

Deputy Editor

Reviewer's Responses to Questions

**Key Review Criteria Required for Acceptance?**

**Methods**

-Are the objectives of the study clearly articulated with a clear testable hypothesis stated?

-Is the study design appropriate to address the stated objectives?

-Is the population clearly described and appropriate for the hypothesis being tested?

-Is the sample size sufficient to ensure adequate power to address the hypothesis being tested?

-Were correct statistical analysis used to support conclusions?

-Are there concerns about ethical or regulatory requirements being met?

Reviewer #1: As stated in the previous review, the methods are appropriate to the aims of the paper.

Reviewer #2: Concerns have been addressed

Reviewer #3: (No Response)

**Results**

-Does the analysis presented match the analysis plan?

-Are the results clearly and completely presented?

-Are the figures (Tables, Images) of sufficient quality for clarity?

Reviewer #1: The results are presented as interviews and focus group extracts, which is fine for the sort of data collected and analysis required. 

Tables: this is a clarification and *not* a suggested revision. Reading the comments of – and responses to – the other reviewers, I would like to clarify that I fully agree with Reviewer 2 and have never stated anything to the contrary. I don’t think similar tables add anything of substance to the paper. I am however fully aware that STEM journals that accept qualitative research tend to apply STEM criteria to it, and that this is where the habit of presenting data this way comes from, but I believe it is something that should be resisted. Table 4, for instance. I assume this table refers to the number of times a topic was mentioned explicitly. But here’s an example of the limited efficacy of tables: on how many occasions was the theme/issued referred to in a mediated manner, and where do we draw the line?

Always on Table 4 (and this *is* a suggested revision): it is unclear whether the ‘factors’ are listed as promoters or consequences of NDTs. Although references to it are made in the text below, it’d be useful if the authors could add a clearer description.

Reviewer #2: Concerns have been addressed

Reviewer #3: (No Response)

**Conclusions**

-Are the conclusions supported by the data presented?

-Are the limitations of analysis clearly described?

-Do the authors discuss how these data can be helpful to advance our understanding of the topic under study?

-Is public health relevance addressed?

Reviewer #1: I think the conclusions could be more precise. The paper presents an interesting case study that has important implications for policy but limits itself to suggesting an 'equity lens' and 'empowerment' to reduce the burden of NDTs. Drawing on the data discussed, I think the authors could suggest more specific ways of using policy towards this aim.

Reviewer #2: Yes

Reviewer #3: (No Response)

**Editorial and Data Presentation Modifications?**

Reviewer #1: This paper needs to be proofread and consistently formatted. The grammar is OK, but the overall text could be set out in a much more compelling way.

Reviewer #2: (No Response)

Reviewer #3: (No Response)

**Summary and General Comments**

Reviewer #1: It feels like the authors want to use one specific context to make broad recommendations, which is not unusual. This aspect of the paper can be found in different sections but it is never too explicit (until lines 550-557), whereas I think it should be the beacon driving the whole writing. Generally, the writing should be a bit tighter 

34-36: Please replace semi-colons with commas

45-46: The meaning of the sentence is unclear. Perhaps the comma between ‘goal’ and ‘shaping’ is a typo.

56: Maybe ‘propagate’ is not the best choice. Worsen? Contribute?

58: ‘as the economies of *some such countries’?

61: saying that NDTs area markers for poverty means equalling the two. NDTs are often found in environments characterised by poverty, and the relationship of exacerbation is certainly mutual, but they are not markers nor proxies.

62: this paper is not a theoretical contribution to knowledge — which is absolutely fine. However, on line 62, I invite the author to rephrase what the paper does. It might be an empirical investigation into X, Y, Z using a political economy approach, for instance, but it does not to ‘extend’ social theory. 

79-80: ‘shift… shifting’. Maybe a synonym can be found to avoid repetition. 

98: NDTs contribute to poverty? The verb ‘to promote’ recalls a form of agency that is not part of the theoretical background of this paper (I think).

207: is there any reason why ‘Namgis’ is between inverted commas? If it is the name of an ethnic group, as suggested by the text, the inverted commas should be removed. 

208-210: the description of the study does not appear to be meaningful. Defining one’s “health and wellbeing in terms of political, economic, social, and cultural aspects of their lives and the ease with which they access natural resources” is a phenomenon common to all social groups. 

211-212: Maybe these could be moved at the beginning of the explanation?

240: ‘Silenced voices’: I find this expression to be a bit problematic. I understand it is a quotation, but it seems to me that qualitative researchers (or most of them) have moved past the idea that they give voice to the silenced (also, by whom? The passive form leaves us guessing for a subject). So, while I agree that qualitative methods offer a deeper and nuanced understanding of phenomena, my suggestion is to avoid opening the ‘can worm’ of the researcher-participant power relationship. Also, I think the reference is incomplete. 

395-397: “The results indicate that people infected with NTDs are unable to take part in communal events due to poverty, shame, and information distortion, which makes them marginalized in social affairs”. It is unclear how poverty related to shame and information distortion, here — and it does, but poverty is a phenomenon much broader than shame (i.e. easier to see how shame would directly relate to NDTs). 

517: please beware punctuation. Maybe “that NTDs… for example, hydrocele is inherited” could be a viable option?

529: the results do not really *demonstrate as in *show. 

539-542: that ‘prioritizing political initiatives […] propagates NDTs’ is factually incorrect, as it depends on the kind of political initiatives (as the authors acknowledge towards the end of the sentence). Please rephrase for accuracy. 

Also, as above, the verb ‘to propagate’ here might have connotations that are too direct and too unique to be applied to the relationship that X has on NDTs. 

550-557: this is possibly the whole point of the paper, which shows that a specific phenomenon (devolution) in a specific context (Kenya) has ignored the issue of NDTs contributing to their exacerbation. Its importance, however, seems to be diluted throughout the paper in favour of more general statements about NDTs and structural poverty. The paper would benefit enormously from having this preceded by a clear statement in the Introduction (e.g. "This paper focuses on post-devolution Kenya to show the political economy of policy initiatives and its influence on NDTs prevalence"). One or two sentences would help us frame the whole paper and the information therein, giving us a straight line to follow the data. 

630-631: ‘Despite our research contributions, our research outlines some limitations as well’. It’s not the research who outlines; it is the research who has/displays some limitations. 

643: the use of which social theory should be clear by now. There is no need to put that between brackets.

Reviewer #2: My concerns have been addressed

Reviewer #3: In my first review I noted that positive elements of this paper are that the topic is important, the paper is well-written, the qualitative fieldwork and data handling seems skillfully done, and the voices of the informants that are presented paint a vivid and detailed picture of the many ways that NTDs harm health and well being in Kenya. 

On the negative side, I was critical because I was unsure about what was learned that is new in this paper. The connections between poverty, poor living conditions, inadequate sanitation, and NTDs, have been well known and understood for a very long time. In a sense the paper restates the basic tenets of “social determinants of health” – that economic, social and political conditions shape population health outcomes. I believe that this is very well established, including with respect to NTDs. 

I still feel that the second critique is valid, as the focus of the paper is on establishing that the social determinants of health contribute to NTDs, which is well known. However, I believe that in this revised version the authors have made their contribution clearer, in part by emphasizing not just issues of poverty, but also how political and governance problems lead to NTDs. In addition, the authors have strengthened the discussion of social and cultural contributors to the harms from NTDs, including stigma and the inaccurate views about the provenance of NTDs. As such, I believe the paper is significantly improved. 

I noted several minor issues which could be corrected in this version: 

Lines 149-150 : please reword this sentence for clarity

Line 567: are political appointees “non state actors?” If they are political appointees they are by definition part of the state apparatus.

PLOS authors have the option to publish the peer review history of their article (what does this mean?). If published, this will include your full peer review and any attached files.

Reviewer #1: No

Reviewer #2: No

Reviewer #3: No
---

## [Decision Letter · Decision Letter 2]

12 Jan 2021

Dear Dr. Ochola,

We are pleased to inform you that your manuscript 'The Impact of Neglected Tropical Diseases (NTDs) on Health and Wellbeing in sub-Saharan Africa (SSA): A Case Study of Kenya' has been provisionally accepted for publication in PLOS Neglected Tropical Diseases.

***

In the proofs, please note the following edits noted by reviewer #1:

Line 30: please change as interviews 'of' key informants *or* 'key-informant interviews'.

Line 46: my initial comment was probably unclear. 'Goal' was a fine term, but there can be no comma separating a verb (‘shapes’, or ‘shaping’ in R1) from the object to which it refers (‘happiness’). Please remove the comma.

Line 99: please consider replacing ‘advance’ with ‘contribute to’. Advance implies a direct and neat relationship, which cannot be inferred given the complexity of the issue.

Lines 407-408: I am not sure why the sentence was rephrased, as ‘leads’ seemed a perfectly fine verb (and better than ‘is a promoter of’).

***

Best regards,

Lisa Dikomitis

Associate Editor

Victoria Brookes

Deputy Editor

Reviewer's Responses to Questions

**Key Review Criteria Required for Acceptance?**

**Methods**

-Are the objectives of the study clearly articulated with a clear testable hypothesis stated?

-Is the study design appropriate to address the stated objectives?

-Is the population clearly described and appropriate for the hypothesis being tested?

-Is the sample size sufficient to ensure adequate power to address the hypothesis being tested?

-Were correct statistical analysis used to support conclusions?

-Are there concerns about ethical or regulatory requirements being met?

Reviewer #1: As previously stated, the methods are adequate; the study design appropriate; and the sample sufficient. Ethical or regulatory requirements were met, and no statistical analysis was required to support conclusions.

Reviewer #2: (No Response)

Reviewer #3: (No Response)

**Results**

-Does the analysis presented match the analysis plan?

-Are the results clearly and completely presented?

-Are the figures (Tables, Images) of sufficient quality for clarity?

Reviewer #1: The analysis matches the plan; the results are clear; the tables of sufficient clarity.

Reviewer #2: (No Response)

Reviewer #3: (No Response)

**Conclusions**

-Are the conclusions supported by the data presented?

-Are the limitations of analysis clearly described?

-Do the authors discuss how these data can be helpful to advance our understanding of the topic under study?

-Is public health relevance addressed?

Reviewer #1: The conclusions support the data presented and the limitations are clearly described. The authors explicitly address the implications of the findings in terms of public health relevance.

Reviewer #2: (No Response)

Reviewer #3: (No Response)

**Editorial and Data Presentation Modifications?**

Reviewer #1: Line 30: please change as interviews 'of' key informants *or* 'key-informant interviews'.

Line 46: my initial comment was probably unclear. 'Goal' was a fine term, but there can be no comma separating a verb (‘shapes’, or ‘shaping’ in R1) from the object to which it refers (‘happiness’). Please remove the comma.

Line 99: please consider replacing ‘advance’ with ‘contribute to’. Advance implies a direct and neat relationship, which cannot be inferred given the complexity of the issue.

Lines 407-408: I am not sure why the sentence was rephrased, as ‘leads’ seemed a perfectly fine verb (and better than ‘is a promoter of’).

Reviewer #2: (No Response)

Reviewer #3: (No Response)

**Summary and General Comments**

Reviewer #1: I believe most comments have been appropriately addressed and the revised manuscript provides a much clearer presentation of its aim, methods, and implications. The additional modifications I suggest are editorial in nature and should be considered minor (if not 'minimal') revisions.

Reviewer #2: My concerns have been addressed.

Reviewer #3: My previous comments have been addressed. I have no further comments on the manuscript.

PLOS authors have the option to publish the peer review history of their article (what does this mean?). If published, this will include your full peer review and any attached files.

Reviewer #1: No

Reviewer #2: No

Reviewer #3: No

---

## [Editor Report · Acceptance letter]

4 Feb 2021

Dear Dr. Ochola,

We are delighted to inform you that your manuscript, " The Impact of Neglected Tropical Diseases (NTDs) on Health and Wellbeing in sub-Saharan Africa (SSA): A Case Study of Kenya ," has been formally accepted for publication in PLOS Neglected Tropical Diseases.

Best regards,

Shaden Kamhawi

co-Editor-in-Chief

Paul Brindley

co-Editor-in-Chief
